# Nanocrystal Preparation of Poorly Water-Soluble Drugs with Low Metal Contamination Using Optimized Bead-Milling Technology

**DOI:** 10.3390/pharmaceutics14122633

**Published:** 2022-11-28

**Authors:** Hironori Tanaka, Yuya Ochii, Yasushi Moroto, Daisuke Hirata, Tetsuharu Ibaraki, Ken-ichi Ogawara

**Affiliations:** 1Formulation R&D Laboratory, Research Division, Shionogi & Co., Ltd., Amagasaki 660-0813, Hyogo, Japan; 2Laboratory of Pharmaceutics, Kobe Pharmaceutical University, Kobe 658-8558, Hyogo, Japan; 3Hiroshima Metal & Machinery Co., Ltd., Hiroshima 737-0144, Hiroshima, Japan

**Keywords:** bead milling, milling parameters, drug nanocrystal, metal contamination, physicochemical properties

## Abstract

Nanocrystal preparation using bead milling is an important technology to enhance the solubility of poorly water-soluble drugs. However, there are safety concerns regarding the metal contaminants generated during bead milling. We have previously reported optimized bead-milling parameters that could minimize metal contamination and demonstrated comparable performance to NanoCrystal^®^, a world-leading contamination-free technology. This study aimed to investigate the applicability of optimized milling parameters for preparing nanocrystals of several poorly water-soluble drugs exhibiting various physicochemical properties. Using our optimized bead-milling parameters, we found that all the tested drugs could be ground into nanosized particles within 360 min. Notably, fenofibrate, which has a low melting point, could be ground into nanosized particles owing to the low level of heat generated during bead milling. Additionally, the concentration of metal contaminants in all the drugs prepared using the optimized milling parameters were approximately ten to twentyfold lower than those prepared without the optimized parameters and were comparable to those prepared using polycarbonate beads, known to minimize metal contamination during bead milling. Our results provide insights into the development of drug nanocrystals with low metal contamination using bead milling.

## 1. Introduction

The development of poorly water-soluble drugs, which often exhibit low oral bioavailability, poses a great challenge in the pharmaceutical industry. Approximately 40–70% of drug candidates are poorly water-soluble [1,2]. Therefore, new solubilization technologies to enhance the solubility of such drugs have been actively investigated [3]. Pharmaceutical solubilization technologies, such as amorphous solid dispersion [4], nanomilling [5], and self-emulsifying drug delivery formulations [6], have been applied from the early to commercial development stages of drug products, and several pharmaceutical products have already been launched using these technologies [7,8].

Among the various solubilization techniques available, nanomilling is one of the most important technologies used to enhance the solubility of poorly water-soluble drugs [9,10]. In nanomilling technology, the drug is maintained in the required crystalline state with reduced particle size, leading to an increased dissolution rate and improved bioavailability [11]. For instance, the oral bioavailability of nanomilled danazol has increased more than thirteenfold compared to that of the unmilled formulation in rats [12]. Notably, nanomilling technology may help address the poor bioavailability of long-acting injectable formulations administered intramuscularly or subcutaneously [13]. Furthermore, drug nanocrystals are applicable to any route of administration, including pulmonary, ophthalmic, and topical drug delivery [11].

To obtain drug nanocrystals, bead milling is the most common technology, requiring high shear-impact forces to achieve high productivity and scalability [14]. In the bead-milling system, drug nanocrystals are milled and dispersed using the collision between grinding beads, such as zirconia beads. However, the collision of zirconia beads results in metal contamination, leading to safety concerns for the final pharmaceutical product [15,16]. In general, a tradeoff correlation exists between milling efficiency and metal contamination; thus, metal contamination generated from bead milling cannot be completely prevented. Therefore, the application of bead milling in the pharmaceutical industry remains limited.

Bead-milling parameters, such as the rotation speed, bead diameter, and bead-filling rate, are key variables in the process [17]. These parameters strongly influence the extent of metal contamination in the bead-milling process [18,19]. Previous studies have predominantly focused on evaluating these parameters to enhance the physical stability of drug nanocrystal suspensions [5]. Although several bead-milling technologies that could reduce metal contamination have been investigated [20,21], suitable alternatives to NanoCrystal^®^ [22,23], recognized as a world-leading contamination-free bead-milling technology, have not yet been developed. To minimize metal contamination, the NanoCrystal^®^ technology uses highly cross-linked polystyrene beads as a grinding medium instead of zirconia beads [22,23].

We have previously reported on the optimized bead-milling parameters that could minimize metal contamination and demonstrated their comparable performance to the NanoCrystal^®^ technology [24]. These optimized parameters were determined using phenytoin as a model drug to reduce metal contamination while maintaining grinding efficiency. However, the relationship between drug physicochemical properties and the critical process parameters of bead milling has not been systematically elucidated [17]. Therefore, whether our optimized milling parameters for minimal contamination could be applied to other poorly water-soluble drugs remains unclear. Thus, this study aimed to investigate the applicability of these optimized milling parameters for minimizing the metal contamination in nanocrystal drug preparation using phenytoin, mefenamic acid, itraconazole, fenofibrate (FNB), and sulfamethoxazole with a wide range of physicochemical properties. Drugs with a high molecular weight, low solubility, and high melting point are processed into a drug nanocrystal suspension using a stabilizer corresponding to their surface energy [25]. Therefore, the wide range of physicochemical properties in these model drugs could affect the applicability of the optimized milling parameters. In addition, drugs with a low melting point, such as FNB, have the potential risk of partial dissolution and being compromised by the frictional heat generated by the collision of the zirconia beads [26]. We also evaluated the particle size transition and metal contamination of each drug after bead milling with the optimized parameters. 

## 2. Materials and Methods

### 2.1. Materials

Phenytoin (PHT) was purchased from Shizuoka Caffeine Industries Co., Ltd. (Shizuoka, Japan). Mefenamic acid (MFA) was purchased from Cheng Fong Chemical Co., Ltd. (Tu Cheng, Taipei). Itraconazole (ITZ) was purchased from J&H Chemical Co., Ltd. (Hangzhou, China). FNB was purchased from Combi-Blocks Inc. (San Diego, CA, USA). Sulfamethoxazole (SMX) was purchased from Virchow Laboratories Ltd. (Telangana, India). Table 1 shows the representative physicochemical properties of these drugs. Polyvinylpyrrolidone (PVP) K-25 and sodium dodecyl sulfate (SDS) were purchased from BASF Japan Ltd. (Tokyo, Japan). Yttria-stabilized zirconia beads (0.3-mm diameter) were purchased from Nikkato Corp. (Osaka, Japan). All other chemicals and solvents were of analytical reagent grade, and purified water was used for the solution preparation.

### 2.2. Procedure for Bead Milling

A previously optimized dispersion medium containing 3% (*w*/*w*) PVP K-25 and 0.25% (*w*/*w*) SDS was used to evaluate the milling time and metal contamination [27]. The drugs were mixed in a dispersion medium to form a suspension using a stirrer set at 500 rpm for 30 min (SM-103, AS ONE Corp., Osaka, Japan). A total of 500 g of the drug suspension was prepared for each drug examined. Wet milling was performed using an Apex-Mill Type-015 (Hiroshima Metal & Machinery Co., Ltd., Tokyo, Japan) as described previously [24]. Briefly, bead milling was performed using the optimized milling parameters for minimal metal contamination, that is, a rotation speed of 2 m/s, bead diameter of 0.3 mm, and bead-filling rate of 75% (*v*/*v*). Yttria-stabilized zirconia beads were used as the grinding medium. All other parameters, such as the flow rate and product temperature, were kept constant. Samples were taken from the outlet of the grinding chamber for particle size measurement and metal contamination determination at different time intervals.

### 2.3. Determination of the Particle Size Distribution in the Drug Nanocrystal Suspensions

The particle size distribution was measured by laser diffraction (LA-950, HORIBA, Ltd., Tokyo, Japan). The measurements were performed using purified water as the diluent. The volumetric median particle size was determined using the refractive index values of 1.61, 1.68, 1.64, 1.64, and 1.55 for PHT, ITZ, SMX, MFA, and FNB, respectively. A refractive index value of 1.33 was set for the measurement medium (water). As the dispersion index, the span value was calculated using the equation below: Span = (D90 − D10)/D50

### 2.4. Determination of the Dissolution Rates of the Drug Nanocrystal Suspensions

Dissolution was tested on an NTR-6400AC (Toyama Sangyo Co., Ltd., Osaka, Japan) using the paddle method. The bath temperature and paddle speed were set at 37.0 °C and 50 rpm, respectively. As a dissolution medium, 900 mL of the second dissolution fluid (pH 6.8; JP2) in the Japanese Pharmacopeia and 1.0% (*w*/*v*) Tween 80 were used. A formulation containing 50 mg PHT was weighed and added to the 900 mL dissolution medium. At 5, 10, 15, 30, 45, and 60 min, 5 mL of the sample was withdrawn and replaced by an equal volume of a fresh medium. All samples were filtered through a 0.45-μm membrane filter (GL Sciences Inc., Tokyo, Japan). The concentration of the PHT in the samples was analyzed using the AQUITY UPLC H-Class System (Waters Corp., Tokyo, Japan) at a detection wavelength of 257 nm. The analytical column was an ACQUITY UPLC BEH C18 (1.7 μm, 2.1 × 150 mm). The mobile phase comprised a 60:40 ratio of 0.1% trifluoroacetic acid and acetonitrile and was delivered at 0.4 mL/min and 35 °C.

### 2.5. Determination of Metal Contamination in the Drug Nanocrystal Suspensions

Metal contamination in the drug nanocrystal suspensions was determined using inductively coupled plasma–mass spectrometry (iCAP Q ICP-MS, Thermo Fisher Scientific, Waltham, MA, USA). Elemental analysis was performed as previously reported [24]. In this study, the unit of metal contamination was μg/mL per volume of suspension.

## 3. Results and Discussion

### 3.1. Reproducibility Validation of the Optimized Milling Parameters in Bead Milling

To confirm the reproducibility of the experiments with the optimized milling parameters, a validation study was performed. Batch-to-batch reproducibility was determined by performing a particle size reduction and metal contamination analysis with the optimized milling parameters using the PHT suspension in triplicate. The detailed validation results are shown in Figure 1 and Table 2. The particle size of PHT (D50 and D90) decreased in a time-dependent manner up to 105 min; subsequently, it may have reached milling equilibrium. The particle size reduction transition during bead milling was similar among the triplicate studies (Figure 1). Additionally, the particle size distribution at <0.2 μm (D50), the target particle size of the drugs after bead milling, showed similar values across the three batches. The standard deviation of D50 and D90 was also small (0.0011 and 0.0065 μm, respectively; Table 2).

Among the various metal elements involved, we focused on the material of zirconia beads and the grinding chamber of the Apex-Mill as possible sources of metal contamination and evaluated the contaminant amounts of zirconium, yttrium, and aluminum after the bead-milling process. Among these elements, the largest metal contamination at <0.2 μm (D50) was observed for zirconium, the main component of the zirconia beads. The total contamination was 1.20–1.38 μg/mL, and no significant differences were observed across the three batches. The resulting standard deviation of the total contamination was also small (0.08 μg/mL; Table 2). In general, bead milling is considered highly reproducible [28]. Therefore, these results demonstrated that our optimized milling parameters were highly reproducible for the particle size reduction transition and metal contamination, validating these parameters.

To confirm the water solubility of the PHT nanocrystals obtained in this study, the dissolution ratio of the PHT suspension before and after bead milling was examined (Figure 2A). The particle size distribution of the PHT suspension used for dissolution testing is shown in Figure 2B. The dissolution ratio of the PHT suspension after bead milling (D50 = 20.06 μm) was approximately twofold higher at 5 min compared with that of the PHT suspension before bead milling (D50 = 0.1998 μm). The PHT nanocrystal suspension reached saturated solubility within 5 min, consistent with the result of a previous study [29]. In addition, the PHT nanocrystal was physically stable for 3 years at 5 °C. These results clearly demonstrated that the drug nanocrystals obtained by bead milling with the optimized parameters were physically stable and improved the dissolution rate of the poorly water-soluble drugs. 

### 3.2. Particle Size after Bead Milling with the Optimized Parameters

Figure 3 shows the particle size transition for each poorly water-soluble drug during bead milling with the optimized parameters. The particle size distribution at <0.2 μm (D50) and the milling time required to reach <0.2 μm (D50) are shown in Table 3. In this study, bead milling was performed for up to 360 min based on the practical manufacturing time per day.

In the optimized milling parameters, the particle size (D50) of all the drugs decreased rapidly within 60 min, gradually reaching milling equilibrium. Although the milling time required to reach <0.2 μm (D50) was different across the tested drugs, all the drugs were ground to <0.2 μm (D50) within 360 min (Figure 3). The milling times required to reach <0.2 μm (D50) for the PHT, MFA, ITZ, FNB, and SMX were 90 min, 90 min, 150 min, 210 min, and 270 min, respectively (Table 3). As for the dispersion index of these drug nanocrystals, the span value was 0.60–0.73, indicating a narrow particle size distribution after bead milling with the optimized parameters. 

Generally, some of the polymers used in bead milling can reduce the interfacial tension and enhance wettability; however, these do not promote surface wetting as effectively as surfactants (e.g., SDS) [5]. According to a previous screening study of nanosuspension formulations, SDS was the most effective in reducing the particle size of cilostazol, compared with other surfactants. In contrast, the particle size of cilostazol was significantly larger in all the suspensions prepared with only polymers. In addition, the combinatorial use of polymers and SDS was also effective in reducing particle size [10]. Given these findings, SDS combined with PVP K-25 was more effective than PVP K-25 alone in terms of enhancing wettability and stabilizing nanoparticles via an electrostatic mechanism during bead milling, leading to the universally successful grinding of all drugs examined to nanosized particles.

Based on Figure 3 and Table 3, together with the physicochemical properties summarized in Table 1, most of the physicochemical properties, including the drug’s intact particle size, were not correlated with the milling time required to reach <0.2 μm (D50). However, except for the SMX, the drugs with a higher melting point were more likely to have a shorter milling time to reach <0.2 μm (D50). These results indicated that the only parameter to have a slight effect on the milling time required to reach the target particle size was the melting point of the drug. Accordingly, a higher melting point was associated with the formation of a more stable nanosuspension with the stabilizers [30]. More detailed studies are required to elucidate the correlation between the physicochemical properties of a drug and the milling time to reach the target particle size. In a future study, we will evaluate the brittle or ductile behavior, the Young’s modulus, the crystal hardness, or any other relevant mechanical parameters to elucidate this correlation.

Owing to the heat generated from milling systems, FNB is difficult to grind into nanosized particles [31]. Nevertheless, our optimized milling parameters successfully ground FNB into nanosized particles, despite its low melting point (83 °C). In this study, the product temperature of the FNB suspension during bead milling was maintained <10 °C, since the bead milling using our optimized milling parameters was performed at an ultra-low speed of 2 m/s, which subsequently placed less stress on the bead-mill system, leading to a reduction in the heat generated during the milling process. Therefore, these results demonstrate that bead milling using our optimized milling parameters can grind various types of poorly water-soluble drugs, including those with a low melting point, into nanosized particles.

### 3.3. Metal Contamination after Bead Milling with the Optimized Parameters

Table 3 summarizes the results of the metal contamination analysis and the milling time required to reach the target particle size (<0.2 μm [D50]), as well as the end of the bead milling (360 min). The highest contamination was observed for zirconium after bead milling. Regarding the milling time required to reach the target particle size without the optimized milling parameters, the contamination in the PHT (120 min) was 10.17 μg/mL. In contrast, with the optimized milling parameters, the zirconium contamination in the PHT (90 min) was 0.64 μg/mL at the target particle size. The SMX (270 min) had the highest contamination (0.98 μg/mL), whereas the ITZ (150 min) had the lowest contamination at 0.35 μg/mL. The zirconium contamination values with the use of the optimized milling parameters were approximately ten- to twentyfold lower than those without optimization. Considering that 220–1400 μg/mL of zirconium contaminant has been reported in a titania nanoparticle slurry prepared using bead milling [18], the amount of metal contaminants generated using our optimized parameters was extremely low. In addition, a patent specification for a bead-milling technology has reported a zirconium contamination of 0.7 μg/mL to obtain materials of 225 nm in size using polycarbonate beads [32]. Even though zirconia beads were used in this study, the zirconium contaminants of the drug nanocrystals obtained with the optimized milling parameters were comparable to those of this patent specification of processing without using zirconia beads. In addition, the risk of microplastic contamination in the bead-milling methods using resin beads, such as Nanocrystal^®^, may be avoided when the new bead-milling technology presented herein is applied.

Regarding the PHT, the total metal contamination was 2.17 μg/mL at the end of bead milling (360 min). The ITZ had the minimum value at 1.47 μg/mL, whereas the MFA had the maximum value at 2.86 μg/mL. There were considerable differences in the contamination values among the tested drugs, despite the constant milling time. These results indicated that the physicochemical property of a drug or suspension influences the metal contamination. We have previously reported that the amount of contamination depends on the pH of the PHT suspension, and the metal contamination eluted from zirconia beads was lower when the suspension pH was closer to a neutral pH [27]. Considering that the ITZ is a weakly basic drug [33] and that the pH of the ITZ suspension was higher and closer to the neutral pH than that of the other drug suspensions (Table 2), this could be why the contaminant level of the ITZ suspension was the lowest.

## 4. Conclusions

We demonstrated that our optimized bead-milling parameters were applicable in the nanocrystal preparation of not only PHT but also various other poorly water-soluble drugs, including those with low melting points. The amount of zirconia contaminants was also demonstrated to be comparable to that of polycarbonate beads, recognized for minimizing metal contaminants during the bead-milling process. These findings may provide useful information in the development of nanocrystal pharmaceutical products with low metal contamination using bead milling. 

## Figures and Tables

**Figure 1 pharmaceutics-14-02633-f001:**
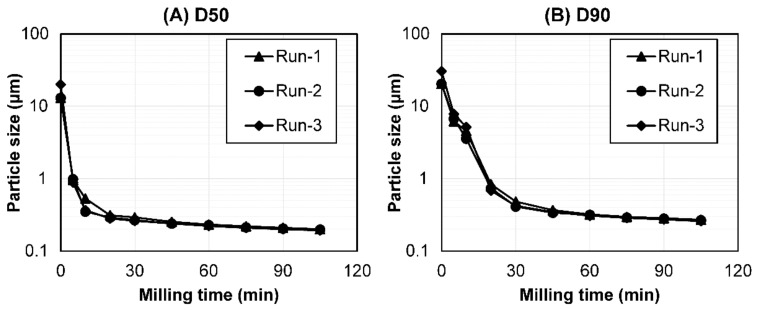
The particle size transition of the PHT as a function of the bead-milling time (*n* = 3). (**A**) D50 and (**B**) D90. Bead milling was carried out for process validation.

**Figure 2 pharmaceutics-14-02633-f002:**
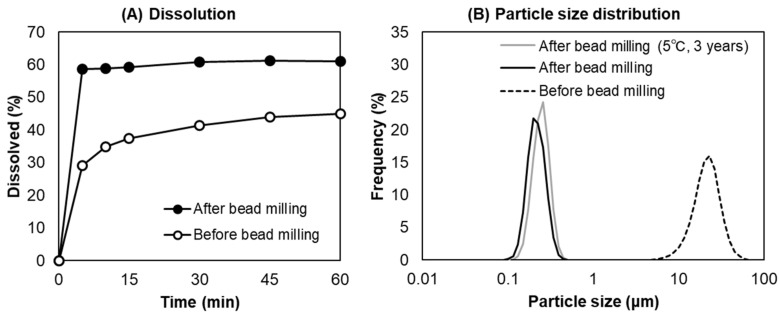
Dissolution ratio of the PHT suspension as a function of the dissolution time (**A**) and the particle size distribution of the PHT suspension before and after bead milling (**B**).

**Figure 3 pharmaceutics-14-02633-f003:**
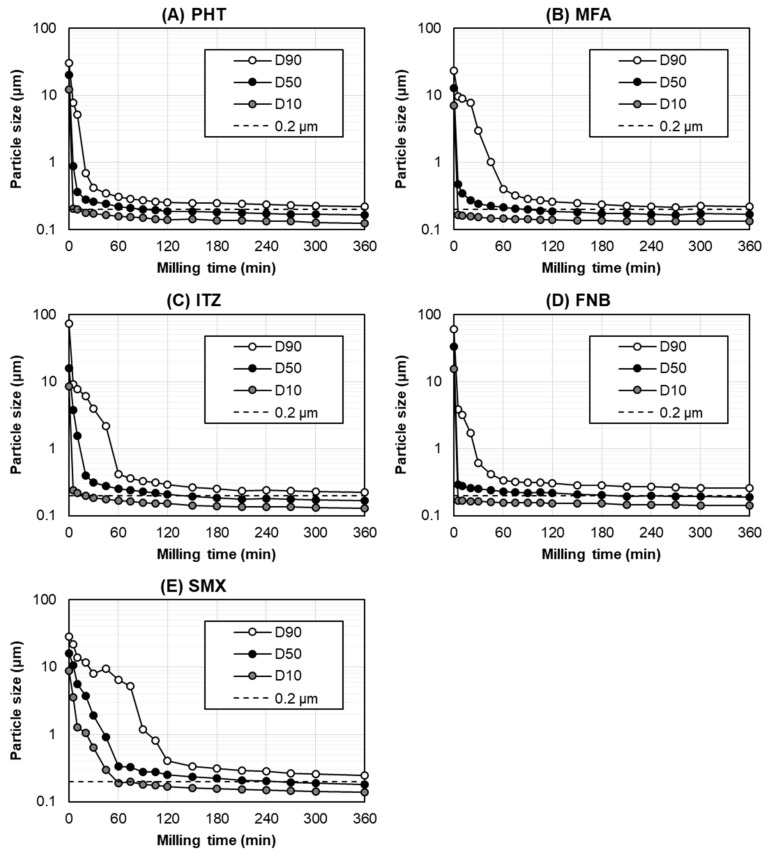
The particle size transition of (**A**) PHT, (**B**) MFA, (**C**) ITZ, (**D**) FNB, and (**E**) SMX as a function of the bead-milling time.

**Table 1 pharmaceutics-14-02633-t001:** The physicochemical properties of poorly water-soluble drugs.

Drug	MW (g/mol) ^1^	Tm (°C) ^1^	Cs (μg/mL) ^2^	Intact Particle Size (μm) ^3^		Suspension pH ^4^
				D50	D90	
PHT	252.27	295	37.8	20.06	30.56	4.17
MFA	241.29	230	36.2	12.88	23.37	4.14
ITZ	705.65	168	0.00964	15.83	73.11	4.61
FNB	360.84	83	0.42	32.84	60.21	4.05
SMX	253.28	168	610	16.12	28.10	4.13

^1^ The Mw and Tm were obtained from the International Journal of Pharmaceutics 538 (2018) 243–249. ^2^ The Cs (water solubility) was obtained from PubChem (http://pubchem.ncbi.nlm.nih.gov/ (accessed on 1 June 2022.)). ^3^ The particle size was measured using Mastersizer 2000. ^4^ Briefly, 5% (*w*/*w*) of the drugs were dissolved in a dispersion medium containing PVP K-25 and SDS.

**Table 2 pharmaceutics-14-02633-t002:** The particle size and metal contamination at the target particle size during the process validation using PHT (*n* = 3).

	Particle Size Distribution				Contaminant Concentration (μg/mL)			
	D10 (μm)	D50 (μm)	D90 (μm)	Span	Zr	Y	Al	Total
Run-1	0.1451	0.1972	0.2575	0.57	0.86	0.39	0.13	1.38
Run-2	0.1453	0.1976	0.2680	0.62	0.69	0.31	0.20	1.20
Run-3	0.1491	0.1998	0.2732	0.62	0.64	0.35	0.24	1.23
Average	0.1465	0.1982	0.2662	0.60	0.73	0.35	0.19	1.27
SD	0.0018	0.0011	0.0065	0.02	0.09	0.03	0.05	0.08

**Table 3 pharmaceutics-14-02633-t003:** Metal contamination after bead milling with the optimized parameters.

Time Point	Milling Parameters	Drug	Milling Time (min)	Particle Size Distribution				Contaminant Concentration (μg/mL)			
				D10 (μm)	D50 (μm)	D90 (μm)	Span	Zr	Y	Al	Total
Target particle size (<0.2 μm)	Not optimized ^1^	PHT	120	0.1504	0.1990	0.2693	0.60	10.17	0.87	2.93	13.97
	Optimized ^2^	PHT	90	0.1491	0.1998	0.2732	0.62	0.64	0.35	0.24	1.23
		MFA	90	0.1419	0.1979	0.2864	0.73	0.71	0.29	0.26	1.26
		ITZ	150	0.1428	0.1932	0.2664	0.64	0.35	0.35	0.36	1.06
		FNB	210	0.1454	0.1955	0.2677	0.63	0.79	0.30	0.26	1.35
		SMX	270	0.1437	0.1931	0.2655	0.63	0.98	0.32	0.24	1.54
End of bead milling	Optimized ^2^	PHT	360	0.1241	0.1627	0.2168	0.57	1.48	0.39	0.30	2.17
		MFA	360	0.1321	0.1686	0.2214	0.53	2.08	0.43	0.35	2.86
		ITZ	360	0.1283	0.1664	0.2208	0.56	0.64	0.42	0.41	1.47
		FNB	360	0.1415	0.1887	0.2553	0.60	1.20	0.35	0.34	1.89
		SMX	360	0.1369	0.1810	0.2463	0.60	1.12	0.32	0.24	1.68

^1^ The rotation speed, bead-filling rate, and bead diameter were set to 4 m/s, 75% (*v*/*v*), and 0.8 mm, respectively. ^2^ The rotation speed, bead-filling rate, and bead diameter were set to 2 m/s, 75% (*v*/*v*), and 0.3 mm, respectively.

## Data Availability

Not applicable.

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
