# Peer review of "Nanocrystal Preparation of Poorly Water-Soluble Drugs with Low Metal Contamination Using Optimized Bead-Milling Technology"

_pharmaceutics, 2022, doi:10.3390/pharmaceutics14122633_

Round 1

Reviewer 2 Report

This work uses bead milling method to obtain the insoluble drug nanocrystals, and the grinding parameters have been optimized to obtain drug nanoparticles with low metal contamination. However, in my opinion, this is a very routine piece of work, which lacks innovation. Therefore, I don’t feel like this work is suitable for publication in the journal of pharmaceutics. Here are some major issues of this work.

1. Bead milling is also a common method to obtain amorphous, the authors need to further characterize the product to determine whether it is nanocrystal or amorphous. In addition, since agglomeration can affect the particle size in a big extent, further characterization should be performed to make sure if the particles get agglomerated after bead milling.

2. As noted in the introduction, this article only evaluated the drug nanocrystal particle size transition and metal contamination. However, there are many other important factors should be considered in preparing drug nanocrystals, such as stability studies and so on. Thus, other additional evaluations should be added to demonstrate the superiority of the obtained nanocrystals.

3. As a research work, the author should at least try to make some reasonable explanations for the experimental phenomena. For example, the author found that the melting point of the drug has a slight effect on the milling time required to reach target particle size, but they did not give any other explanations. As a scientific researcher, I think we should at least try to give some possible mechanisms, instead of waiting for later people to do it.

Reviewer 3 Report

The authors attempt to prove the generality of their optimized process parameters, by applying them on several poorly soluble drugs of different physicochemical properties. The design of the study however, seems to ignore some important properties, and especially the mechanical strength of the crystals. According to Table 2, MW, melting point and saturation solubility were chosen as the variables differentiating the various substances used in this study, while it would be more meaningful to choose them on the basis of their mechanical properties. Brittle or ductile behavior, Young modulus, crystal hardness or any relevant mechanical parameter would be more useful than the melting point or solubility in order to draw general conclusions about the optimized milling process.

Also, it is not clear why the optimization carried out on phenytoin, in reference 24, would be valid for substances with different mechanical properties. There is no reason to accept that the results presented in this study are the best possible (optimal) and no further improvements are possible.

The authors should also discuss how it is possible that PVP K25, which isn't such a successful stabilizer considering the relevant literature, seems to be universally successful in this study. A discussion considering the surface chemistry of the tested substances in relation to the PVP's chemistry would be useful.

Finally, the phrase "drugs, we expect that our proposed approach will be applicable to peptide drugs, which are vulnerable to physical stress" (lines 250-251) of the conclusions, is highly speculative, and is not a conclusion based on data of this study anyway.

Round 2

Reviewer 2 Report

This manuscript can be accepted in present form. 

Reviewer 3 Report

The authors have made an effort to provide the necessary clarifications to meet this referee's points. The manuscript is substantially improved.